# Leptin, Ghrelin, and Leptin/Ghrelin Ratio in Critically Ill Patients

**DOI:** 10.3390/nu12010036

**Published:** 2019-12-21

**Authors:** Yaseen M. Arabi, Dunia Jawdat, Hasan M. Al-Dorzi, Hani Tamim, Waleed Tamimi, Abderrezak Bouchama, Musharaf Sadat, Lara Afesh, Mashan L. Abdullah, Walid Mashaqbeh, Maram Sakhija, Abdulaziz Al-Dawood

**Affiliations:** 1Intensive Care Department, King Abdulaziz Medical City, Riyadh 11426, Saudi Arabia; aldorziha@ngha.med.sa (H.M.A.-D.); bouchamaab@ngha.med.sa (A.B.); sadatmu@ngha.med.sa (M.S.); sakkijham@ngha.med.sa (M.S.); dawooda@ngha.med.sa (A.A.-D.); 2College of Medicine, King Saud bin Abdulaziz University for Health Sciences, Riyadh 11426, Saudi Arabia; jawdatd@ngha.med.sa (D.J.); hani_t@hotmail.com (H.T.); tamimiw@ngha.med.sa (W.T.); aldlamyme@ngha.med.sa (M.L.A.); mashaqbehw@ngha.med.sa (W.M.); 3King Abdullah International Medical Research Center, Riyadh 11426, Saudi Arabia; afeshla@ngha.med.sa; 4Cord Blood Bank, King Abdullah International Medical Research Center, Riyadh 11426, Saudi Arabia; 5Department of Internal Medicine, American University of Beirut-Medical Center, Beirut 1107 2020, Lebanon; 6Department of Clinical Laboratory, King Abdulaziz Medical City, Riyadh 11426, Saudi Arabia; 7Department of Experimental Medicine, King Abdullah International Medical Research Center, Riyadh 11426, Saudi Arabia

**Keywords:** critical illness, adipokines, obesity, BMI, energy restriction, inflammatory markers

## Abstract

The objective of this study was to evaluate leptin, ghrelin, and leptin/ghrelin ratio in critically ill patients and association of leptin/ghrelin ratio with outcomes. This is a sub-study of the PermiT trial (ISRCTN68144998). A subset of 72 patients who were expected to stay >14 days in the Intensive care unit were enrolled. Blood samples were collected on days 1, 3, 5, 7, and 14. Samples were analyzed for leptin and active ghrelin in addition to other hormones. Baseline leptin/ghrelin ratio was calculated, and patients were stratified into low and high leptin/ghrelin ratio based on the median value of 236. There was a considerable variation in baseline leptin level: Median 5.22 ng/mL (Q1, Q3: 1.26, 17.60). Ghrelin level was generally low: 10.61 pg/mL (Q1, Q3: 8.62, 25.36). Patients with high leptin/ghrelin ratio compared to patients with low leptin/ghrelin ratio were older, had higher body mass index and more likely to be diabetic. There were no differences in leptin/ghrelin ratio between patients who received permissive underfeeding and standard feeding. Multivariable logistic regression analysis showed that age and body mass index were significant independent predictors of high leptin–ghrelin ratio. Leptin–ghrelin ratio was not associated with 90-day mortality or other outcomes. Age and body mass index are predictors of high leptin/ghrelin ratio. Leptin/ghrelin ratio is not affected by permissive underfeeding and is not associated with mortality.

## 1. Introduction

Leptin and ghrelin are two orexitropic hormones with opposite effects on energy homeostasis. Leptin is an adipokine that is released by the adipose tissue. It regulates food intake and energy expenditure and suppresses appetite (satiety hormone) by sending signals about the peripheral obesity to the central nervous system [1]. Leptin has other effects on glucose homeostasis, immune response, growth and differentiation and angiogenesis and have been implicated in the pathogenesis of hypertension, atherosclerosis and cancer. During critical illness, leptin levels are high initially and then decrease over time [2,3,4], suggesting that it may play a role in the pathogenesis and/or outcome of disease [5]. On the other hand, ghrelin is secreted in the stomach [6], and is a potent appetite stimulator (hunger hormone). Ghrelin increases during fasting in normal subjects [7], helps in short-term and long-term weight regulation and usually promotes weight gain and adiposity [8]. It has been shown to have cardiovascular and anti-inflammatory effects [9,10,11]. Ghrelin was found to mediate improvement of tissue perfusion in severe sepsis [12], and down-regulation of proinflammatory cytokines in sepsis through activation of the vagus nerve [13]. A study showed that ghrelin levels are reduced in intensive care unit (ICU) patients which seemed to be responsible for suppression of appetite and nutritional intake and gastrointestinal dysfunction including delayed gastric emptying [14].

Feeding dose and composition affects leptin and ghrelin inversely [15]. Energy deficit causes a rapid initial decrease in circulating leptin levels that becomes more marked with progressive loss of body fat [16], and a marked increase in ghrelin. High protein meals increase circulating concentrations of the gut hormones PYY and GLP-1 and decrease ghrelin concentration [17]. An increase in dietary protein from 15% to 30% of energy at a constant carbohydrate intake produces a sustained decrease in ad libitum caloric intake that is probably mediated by increased central nervous system leptin sensitivity and results in significant weight loss [18]. This anorexic effect of protein may contribute to the weight loss produced by low-carbohydrate diets [18]. As there is no consistent relation between both leptin and ghrelin and weight loss and then subsequent weight regain [19], the leptin/ghrelin ratio has been suggested to be a better marker and a predictor of energy restriction treatment success or failure [19]. In one study, higher baseline (week 0) and after treatment (week 8) leptin/ghrelin ratio was associated with an increased risk for weight regain in an 8-week hypocaloric diet program [20]. In a recent study which evaluated the leptin/ghrelin ratio in a fasting state and after the intake of meals with varying macronutrient contents, there was a significant difference in postprandial leptin/ghrelin ratio in normal body weight and overweight/obese men [21].

There is limited information about leptin, ghrelin, and the leptin/ghrelin ratio in critically ill patients and in association with feeding. The objective of this study was to evaluate leptin, ghrelin, and the leptin/ghrelin ratio in critically ill patients and the association of leptin/ghrelin ratio with outcomes.

## 2. Methods

This is a sub-study of the PermiT (Permissive Underfeeding versus Target Enteral Feeding in Adult Critically Ill Patients, Current Controlled Trials number, ISRCTN68144998) trial in which the critically ill patients were randomized to permissive underfeeding (40–60% of calculated caloric requirements) or standard feeding (70–100%) for up to 14 days while maintaining similar protein intake in both groups [22]. Caloric intake was calculated using the Penn State equation for mechanically ventilated patients with body mass index (BMI) ≤30 and Ireton-Jones equation for those with BMI ≥30. Protein target was maintained uniformly at 1.2 to 1.5 g per kg of body weight per day with additional protein (Resource Beneprotein, Nestle Healthcare, Lausanne, Switzerland) provided as needed. The trial found no difference between the two groups in the primary endpoint of 90-day mortality (relative risk 0.94, 95% confidence interval 0.76, 1.16, *p* = 0.58). In this sub-study, we enrolled a subset of patients who were expected to stay >14 days in the ICU at King Abdulaziz Medical City-Riyadh and consented for blood sample collection. The sub-study was approved by Institutional Board Review of the Ministry of the National Guard Health Affairs, Riyadh, Saudi Arabia. Blood samples were collected on days 1, 3, 5, 7, and 14. Samples were centrifuged for 20 min and stored at −80°C.

### 2.1. Laboratory Measurements

Samples were analyzed using Millipore (Merck Mellipore, Darmstadt, Germany) with Luminex 3D platform (Luminex, Austin, TX, USA) for leptin, active ghrelin as well as the following adipokines (adiponectin, monocyte chemotactic protein-1 (MCP-1), resistin, adipsin, and plasminogen activator inhibitor-1 (PAI-1)), pancreatic hormones (amylin, C-peptide, glucagon, pancreatic polypeptide (PP)), gut hormones (gastric inhibitory polypeptide (GIP), glucagon-like peptide (GLP), polypeptide YY), glycoproteins (neutrophil gelatinase associated lipocalin or NGAL (lipocalin-2 NGAL) and selected inflammatory markers (interleukin-1B, interleukin-6, and tumor necrosis factor-α). We did not add serine protease inhibitor to blood samples after collection.

Reported normal leptin levels are median 4.0 (Q1, Q3: 1.7, 7.2) ng/mL [23]. Reported active ghrelin in lean subjects mean 411.8 pg/mL (SD 57.4) and 180.4 pg/mL (SD 18.5) in obese subjects [24]. Baseline leptin and active ghrelin levels were evaluated and the leptin/ghrelin ratio was calculated as leptin in ng/mL multiplied by 10^3^ and divided by ghrelin in pg/mL [20]. The median leptin/ghrelin ratio for this study cohort was 236, which was used as a cut off and the patients with leptin/ghrelin ratio ≤236 were considered to have low leptin/ghrelin ratio and the patients with ratio >236 were considered to have high leptin/ghrelin ratio.

### 2.2. Data Collection

We collected the baseline data which included demographics, physiological parameters (Acute Physiology and Chronic Health Evaluation Scores (APACHE) II, Sequential Organ Failure Assessment (SOFA) score, the ratio of partial pressure of arterial oxygen to the fraction of inspired oxygen (PaO2:FiO2), Glasgow coma scale and laboratory parameters. The nutritional data were collected for the period of intervention which was 14 days and included the average caloric intake and average protein intake. In addition, data about cointerventions including insulin dose, daily blood glucose and other medications were collected. Mortality outcome at 28, 90, and 180 days was collected. Other outcomes included hospital and ICU mortality, new renal replacement therapy, ICU-associated infections, ICU and hospital length of stay (LOS) and mechanical ventilation duration.

### 2.3. Statistical Analysis

Categorical variables were reported as frequencies with percentages and continuous variables as medians with quartiles 1 and 3 (Q1, Q3). Categorical variables were compared using chi-square or Fisher’s exact test and continuous variables using Mann–Whitney U test based on non-normality assumption and small sample size. Multivariate regression analysis was carried out to assess the predictors of high leptin/ghrelin ratio adjusting for selected baseline variables of clinical interest (age, BMI, sex, diabetes, sepsis and APACHE II). We also carried out logistic regression models to examine the association between leptin/ghrelin ratio and 90-day mortality adjusting for the same variables. The results were presented as adjusted odds ratio (OR) with 95% confidence interval (CI). To assess whether low leptin/ghrelin ratio compared to high leptin/ghrelin ratio was associated with levels of different hormones and other inflammatory markers over time, we constructed repeated measure mixed linear model. *p* value of ≤0.05 was considered significant. All statistical analyses were performed using SAS version 9.2 (SAS Institute, Cary, NC, USA).

## 3. Results

Seventy-two patients were included in this study and their characteristics were described in Table 1 according to the leptin/ghrelin ratio. There was a considerable variation in baseline leptin level; median 5.22 ng/mL (Q1, Q3: 1.26, 17.60), while active ghrelin level was generally low: 10.61 pg/mL (Q1, Q3: 8.62, 25.36). Patients with high leptin/ghrelin ratio were older, obese and more likely to be diabetic. The total daily caloric intake and protein intake was similar in the two groups. However, more patients in the high leptin–ghrelin ratio group were more likely to be on aspirin and statins in comparison to the low leptin–ghrelin ratio group (16 (44.4%) vs. 7 (19.4%), *p* = 0.02 and 19 (52.8%) vs. 8 (22.2%), *p* = 0.007), respectively (Table 2).

Appendix A also shows no differences in the serial leptin/ghrelin ratio between the permissive underfeeding and standard feeding groups during the intervention period.

### Relationship between Leptin–Ghrelin Ratio and Other Hormones

Figure 1, Appendix A and Appendix A describe the results of repeat measure analysis using the mixed linear model of the differences in hormones between the high and low leptin/ghrelin ratio groups. With the exception of glucagon levels, which were lower in the high leptin/ghrelin ratio group over time (*p* = 0.007), other hormones were not different.

## 4. Outcomes

There was no significant difference in all-cause 90-day mortality between the two groups. There was also no significant difference in other outcomes including length of stay in the ICU or in the hospital, ventilation duration, ICU-associated infections and new renal replacement therapy between the two groups except ventilator associated pneumonia which was more common in patients with low leptin/ghrelin ratio group in comparison to the high leptin/ghrelin ratio group (14 (38.9%) vs. 8 (22.2%), *p* = 0.04). Multivariable logistic regression analysis showed no significant association between high leptin/ghrelin ratio and 90-day mortality (adjusted OR: 0.78, 95% CI: 0.18, 3.35; *p* = 0.73) or any other study outcomes (Table 3).

Stepwise multivariable logistic regression analysis showed that age and BMI were independent predictors of high leptin–ghrelin ratio (Age: aOR for each 1-year increase, 1.03 95% CI 1.002, 1.06; *p* value= 0.04); BMI: aOR for each 1-unit increase 1.22 95% CI, 1.09, 1.37; *p* = 0.0004) (Table 4).

## 5. Discussion

In this cohort of critically ill patients, we found large variations in leptin levels while active ghrelin levels were low. The low active ghrelin level in critically ill patients is in line with other previous studies and probably contributes the suppressed appetite and gastroparesis [14,15,16,17,18,19,20,21,22,23,24,25].

In the current study, BMI was the only predictor of the leptin/ghrelin ratio. This finding is consistent with the fact that leptin, which is anorexigenic, is secreted by the adipose tissue and that more leptin is excreted by the obese. Most obese individuals and subjects with a predisposition to regain weight after losing it have higher leptin concentrations than lean individuals [26]. However, appetite is not effectively suppressed in these individuals suggesting leptin resistance [26].

Fasting in healthy subjects usually suppresses leptin and increases ghrelin secretion. Using sensitive radio-immunoassays, a study found that plasma ghrelin and leptin were secreted in pulsatile fashion in rats consuming ad libitum food [27]. Fasting augmented all parameters of ghrelin pulsatile secretion and diminished leptin secretion by selectively attenuating the pulse amplitude [27]. In the TICASOS trial, in which ICU patients were randomized to receive nutrition with an energy target determined either by repeated indirect calorimetry measurements (study group, n = 47) or a weight-based formula (25 Kcal/kg/day; control group, n = 44), there were no significant between-group differences in serum leptin, ghrelin and other adipokines (resistin and adiponectin) on day 1 or day 7 [28]. In the control group, serum ghrelin increased significantly over time (*p* < 0.05) [28]. For the whole group, a more positive cumulative energy balance and a lower maximal negative energy balance were associated with a significantly smaller increase in serum ghrelin levels (*p* = 0.008 and *p* = 0.035, respectively) [28]. Caloric restriction in healthy individuals is expected to be associated with lower leptin/ghrelin ratio. However, in the current study, we found that in critically ill patients the leptin/ghrelin ratio over time was not affected by moderate caloric restriction.

Leptin and ghrelin may have important roles in critical illness. High leptin levels have been observed upon ICU admission, with levels subsequently decreasing [2,3]. In mouse endotoxemia and cecal ligation puncture models of sepsis, elevated levels of leptin and soluble leptin receptor have been observed [5]. Additionally, exogenously administered leptin was associated with increased expression of adhesion and coagulation molecules, macrophage infiltration into the liver and kidney, and endothelial barrier dysfunction increased and with increased mortality [5]. However, in critically ill patients, ghrelin levels are reduced despite low nutritional intake. Plasma ghrelin levels are associated with systemic inflammation, and ghrelin has potent inhibitory effects on proinflammatory cytokines including IL-1β, IL-6, and TNF-α. Intraperitoneal administration of ghrelin (23 μg/kg/d) or saline after a colonic anastomosis was associated with beneficial anti-inflammatory and antioxidant effects [10]. In rats with hemorrhagic shock-induced ALI, exogenous ghrelin was associated with less lung injury and attenuated the inflammatory response [29]. Another study found that exogenous ghrelin attenuated the inflammatory response and mortality after whole body irradiation in rats [30]. In critically ill patients, we did not find mortality difference between the low and high leptin/ghrelin groups. On multivariable logistic regression analysis, leptin/ghrelin ratio was also not associated with 90-day mortality or other outcomes.

The study results should be interpreted in the light of its strengths and limitations. A strength is that data came from a randomized controlled trial and that we measured the levels of a large number of adipokines at multiple time intervals. The limitations include the relatively small sample size, which makes the study underpowered to detect a mortality difference and that the study included patients who were expected to stay in the ICU for ≥14 days, which may affect the generalizability of study results. We did not add serine protease inhibitor to blood samples after collection, which may have resulted in underestimation of ghrelin levels. Because of the sample size, we were unable to perform subgroup analyses. 

In conclusion, there is a considerable variation in baseline leptin level in critically ill patients, while active ghrelin level is generally low. Age and BMI are predictors of high leptin/ghrelin ratio. Leptin/ghrelin ratio is not affected by permissive underfeeding and is not associated with mortality.

## Figures and Tables

**Figure 1 nutrients-12-00036-f001:**
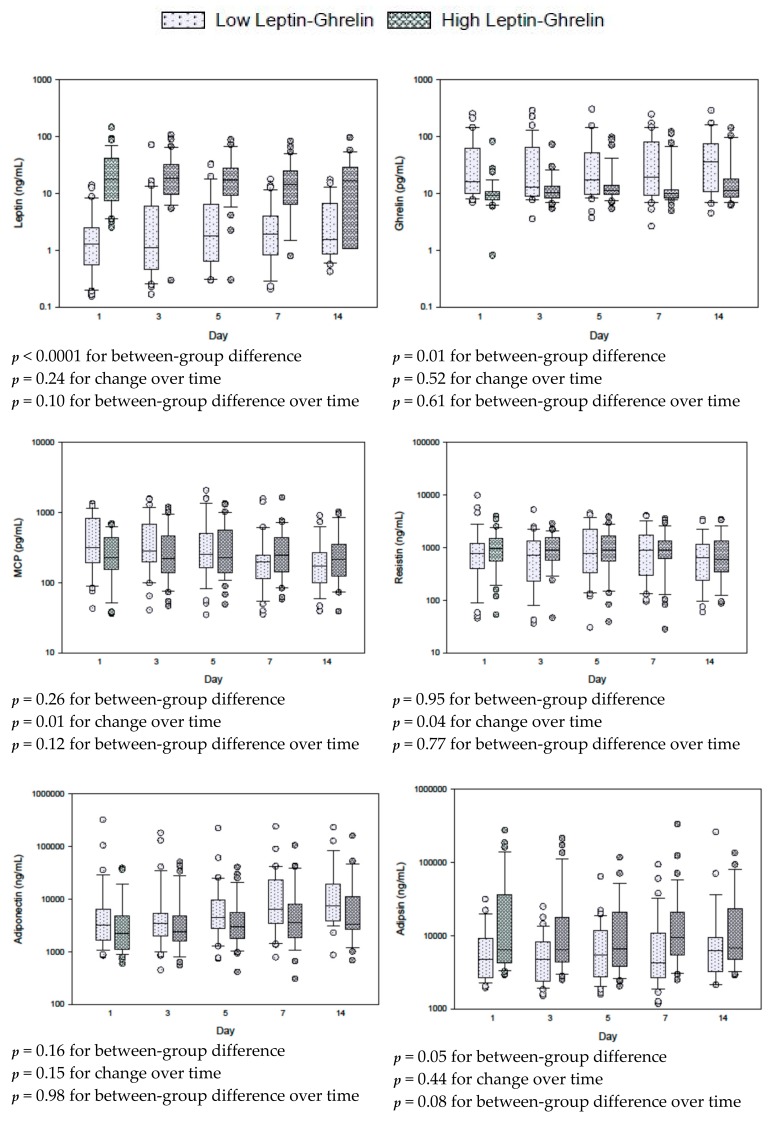
Serial measurements for leptin, ghrelin and other adipokines in patients with low and high leptin/ghrelin ratio at baseline. The differences between groups, with time and between groups with time (group × time) were tested by repeated measures mixed linear models. Box plots are displayed with medians and quartiles 1 and 3. The error bars refer to 10th and 90th percentiles.

**Table 1 nutrients-12-00036-t001:** Baseline characteristics of patients low and high leptin–ghrelin ratio.

Variables	Low Leptin/Ghrelin Ratio Group * (≤236 pg/mL) *n* = 36	High Leptin/Ghrelin Ratio Group (>236 pg/mL) *n* = 36	*p* Value
Age—years median (Q1, Q3)	30.6 (24.0, 63.9)	64.3 (46.7, 72.2)	0.002
Female sex—no (%)	7 (19.4)	13 (36.1)	0.11
BMI—kg/m^2^ median (Q1, Q3)	23.7 (21.0, 26.5)	31.2 (27.6, 36.0)	<0.0001
Admission category—no (%)			
Medical	17 (47.2)	23 (63.9)	0.30
Surgical	4 (11.1)	4 (11.1)	
No-operative trauma	15 (41.7)	9 (25.0)	
Diabetes—no (%)	11 (30.6)	26 (72.2)	0.0004
Sepsis—no (%)	6 (16.7)	11 (30.6)	0.17
Traumatic brain injury—no (%)	1 (2.8)	2 (5.6)	0.56
APACHE II score—median (Q1, Q3)	21 (12, 26)	21.0 (14.0, 26.5)	0.71
Glasgow Coma Scale—median (Q1, Q3)	3 (3, 4)	3 (3, 7)	0.18
SOFA Score on Day 1—median (Q1, Q3)	11 (9, 12)	10 (8, 13)	0.62
Mechanical ventilation—no (%)	34 (94.4)	33 (91.7)	0.64
Vasopressor therapy—no (%)	26 (72.2)	21 (58.3)	0.22
Renal replacement therapy—no (%)	1 (2.8)	2 (5.6)	0.56
Inclusion blood glucose—(mmol/L) median (Q1, Q3)	8.4 (6.7, 12.7)	11.7 (8.4, 14.0)	0.05
Creatinine—(µmol/L) median (Q1, Q3)	81 (67.5, 114.5)	95 (75.5, 118.0)	0.12
Bilirubin—(µmol/L) median (Q1, Q3)	17.8 (9.8, 29.4)	11.8 (8.6, 26.0)	0.18
Platelets—(10^9^/L) median (Q1, Q3)	170.5 (137, 230)	197 (138, 267)	0.38
INR—median (Q1, Q3) *	1.2 (1.0, 1.3)	1.1 (1.0, 1.2)	0.29
SOFA hypotension score—median (Q1, Q3)	4 (3, 4)	3 (1.5, 4.0)	0.39
PaO_2_/FiO_2_ ratio—median (Q1, Q3)	138 (118, 281.5)	150 (68, 242)	0.40
HbA1c—median (Q1, Q3)	0.06 (0.05, 0.07)	0.07 (0.06, 0.08)	0.10
C-reactive protein—(mg/L) median (Q1, Q3)	126.5 (68.1, 194.5)	108 (47.9, 160.0)	0.35
Albumin—(g/L) median (Q1, Q3)	28 (26, 31)	28 (28, 33)	0.51
Pre-albumin—(g/L) median (Q1, Q3)	0.12 (0.09, 0.15)	0.11 (0.09, 0.13)	0.41
24 h urinary nitrogen—(mmol/d) median (Q1, Q3)	209 (131, 326)	266.5 (173, 391.0)	0.10
Transferrin—(g/L) median (Q1, Q3)	1.3 (0.9, 1.5)	1.4 (1.1, 1.8)	0.14
Hemoglobin—(g/L) (median (Q1, Q3)	120 (92.0, 126.0)	105 (90, 131)	0.88
Serum lipid levels—(mmol/L) median (Q1, Q3)			
Triglycerides	1.1 (0.7, 1.8)	1.2 (0.9, 1.8)	0.72
Total cholesterol	2.5 (2.0, 2.9)	2.8 (2.0, 3.3)	0.32
Low-density lipoprotein	1.0 (0.6, 1.3)	1.2 (0.8, 1.7)	0.22
High-density lipoprotei	0.6 (0.3, 0.9)	0.6 (0.5, 0.9)	0.40
Minute ventilation—(L/min) median (Q1, Q3)	8.5 (7.1, 10.9)	9.4 (7.9, 11.5)	0.24
Maximum temperature—Celsius median (Q1, Q3)	37.0 (36.6, 37.0)	37.1 (36.7, 37.6)	0.82
Baseline leptin level—(ng/mL)	1.26 (0.57, 2.43)	17.59 (7.73, 40.10)	<0.0001
Baseline active ghrelin level—(pg/mL)	16.2 (10.1, 59.2)	9.2 (7.6, 10.8)	<0.0001
Baseline leptin/ghrelin ratio *	53.4 (22.2, 119.9)	1747.8 (736.2, 4507.5)	<0.0001

APACHE II: Acute Physiology and Chronic Health Evaluation II; BMI: body mass index; INR: international normalized ratio; Q1, Q2: first and third quartiles; SOFA: Sequential Organ Failure Assessment; PaO_2_:FIO_2_ ratio: the ratio of partial pressure of oxygen to the fraction of inspired oxygen; SD: standard deviation. Continuous variables were compared using Wilcoxon–Mann–Whitney test and categorical variables using chi-square test. * Leptin–ghrelin ratio was calculated as leptin in ng/mL multiplied by 10^3^ and divided by active ghrelin in pg/mL.

**Table 2 nutrients-12-00036-t002:** Daily caloric intake, protein intake, insulin and glucose data in patients with low and high leptin–ghrelin ratio.

Variable	Low Leptin/Ghrelin Ratio Group (≤236) *n* = 35	High Leptin/Ghrelin Ratio Group (>236) *n* = 34	*p* Value
**Feeding group allocation**			
Permissive underfeeding—no (%)	22 (61.1)	13 (36.1)	0.03
Standard feeding—no (%)	14 (28.9)	23 (63.9)	
**Nutritional data**			
Calculated caloric requirement—(kcal/day) median (Q1, Q3)	1675 (1475, 1945)	1860.5 (1683, 2074)	0.06
Study caloric target—(kcal/day) median (Q1, Q3)	1193 (988, 1620)	1672(1206, 1995)	0.009
Achieved daily caloric intake (kcal) median (Q1, Q3)	957 (787, 1177)	1090 (803, 1607)	0.12
% of requirement achieved—median (Q1, Q3)	57(45, 63)	57 (49, 87)	0.41
**Caloric source**—(kcal) median (Q1, Q3)			
Enteral	824 (618, 1115)	998 (727, 1489)	0.08
Propofol	72 (17, 120)	79 (23, 149)	0.58
Dextrose	4 (0, 43)	0 (0, 14)	0.10
Parental nutrition	0 (0, 0)	0 (0, 0)	0.33
Calculated protein requirement—(g/day) median (Q1, Q3)	78 (66, 89)	85 (73, 95)	0.13
Achieved protein intake—(g/day) median (Q1, Q3)	51 (41, 62)	56 (39, 70)	0.23
% of requirement achieved—median (Q1, Q3)	67 (59, 82)	74 (50, 84)	0.52
**Protein Source**—(g/day) median (Q1, Q3)			
Main enteral formula	31 (23, 47)	38 (28, 56)	0.08
Supplemental enteral protein	16 (0, 22)	4 (0, 27)	0.59
Received insulin—no (%)	16 (44.4)	28 (77.8)	0.004
Daily insulin dose— (unit) median (Q1, Q3)	0.0 (0.0, 6.7)	22.2 (2.8, 47.4)	0.0003
**Medications**—no (%)			
Beta-blockers	17 (47.2)	17 (47.2)	1.00
Aspirin	7 (19.4	16 (44.4)	0.02
Angiotensin-converting enzyme inhibitors	4 (11.1)	5 (13.9)	0.72
Angiotensin II receptor blockers	0	1 (2.8)	0.31
Statins	8 (22.2)	19 (52.8)	0.007
**Formulae**—no (%)			
Disease non-specific	19 (52.8)	12 (33.3)	0.096
Disease specific	17 (47.2)	24 (66.7)	
Duration of feeding interruption per day—hours median (Q1, Q3)	3.9 (2.2, 5.8)	4.0 (2.1, 6.0)	0.95
Fluid intake—ml/day median (Q1, Q3)	3354.6 (2549.9, 4390.8)	2804.1 (2231.3, 3863.5)	0.14
Fluid output—ml/day median (Q1, Q3)	3248.3 (2222.0, 3963.0)	2743.0 (2171.4, 3574.4)	0.36
Blood glucose—mmol/L median (Q1, Q3)	7.2 (6.2, 8.8)	10.2 (7.2, 12.0)	0.0008

Disease-non-specific formula: Osmolite, Jevity, Promote, Ensure plus, Resourse, Ensure, Resource plus, Jevity (1.2); Disease specific formula: Glucerna, Nutric hepatic, Nepro, Pulmocare, Novasource Renal, Peptamen (1.0), Peptamen (1.2), Suplena, Oxepa; Continuous variables were compared using Wilcoxon–Mann–Whitney test and categorical variables using chi-square test.

**Table 3 nutrients-12-00036-t003:** Outcome data of patients with low and high leptin–ghrelin ratio.

Outcomes	Low Leptin–Ghrelin Ratio Group (≤236) *n* = 35	High Leptin–Ghrelin Ratio Group (>236) *n* = 34	*p*-Value	Adjusted Odds Ratio (95% CI) *	Adjusted *p*-Value
**28-day mortality—no (%)**	4 (11.1)	6 (16.7)	0.50	1.76 (0.34,8.94)	0.50
**90-day mortality—no (%)**	7 (19.4)	6 (16.7)	0.77	0.78 (0.18,3.35)	0.73
**180-day mortality—no (%)**	7 (19.4)	8 (22.2)	0.77	1.12 (0.27,4.65)	0.88
**ICU mortality—no (%)**	4 (11.1)	3 (8.3)	0.69	0.61 (0.09,4.40)	0.62
**Hospital mortality-no (%)**	5 (13.9)	6 (16.7)	0.74	1.49 (0.31,7.07)	0.62
**ICU LOS—(days) median (Q1, Q3)**	16 (10.5, 21.5)	15 (9, 20.5)	0.84		
**Hospital LOS—(days) median (Q1, Q3)**	39.5 (18, 90.5)	35 (20.5, 58.5)	0.32		
**Ventilation duration—(days) median (Q1, Q3)**	10 (7, 15.5)	11.0 (5, 17.5)	0.38		
**New renal replacement therapy—** **no (%)**	3 (8.8)	8 (23.5)	0.10	1.62 (0.28,9.51)	0.59
**Urinary tract infection—no (%)**	5 (13.9)	8 (22.2)	0.36	0.84 (0.18, 4.01) (0.52,6.05)	0.83
**Ventilator-associated pneumonia—no (%)**	14 (38.9)	8 (22.2)	0.12	0.20 (0.04, 0.89)	0.04
**Feeding intolerance—no (%)**	8 (22.2)	11 (30.6)	0.42	1.16 (0.29, 4.67)	0.84
**Diarrhea-no (%)**	13 (36.1)	11 (30.6)	0.62	0.47 (0.12, 1.80)	0.27
**High gastric residual—no (%)**	5 (13.9)	6 (16.7)	0.74	0.70 (0.12, 4.00)	0.69

CI: confidence interval; ICU: intensive care unit, LOS: length of stay; Q1, Q3: first and third quartiles; Feeding intolerance was defined as vomiting, abdominal distention, or a gastric residual volume of more than 200 mL. Diarrhea was defined as three or more loose or liquid stools per day for 2 consecutive days. High gastric residual: gastric residual volume of more than 500 mL. Continuous variables were compared using Wilcoxon–Mann–Whitney test and categorical variables using chi-square test. * adjusted for age, BMI, sex, diabetes, sepsis and APACHE II.

**Table 4 nutrients-12-00036-t004:** Predictors of high leptin–ghrelin ratio using stepwise multivariable logistic regression analysis. The following variables were entered in the model: Age, BMI, sex, diabetes, sepsis, and APACHE II.

Variables	OR (95% CI)	*p* Value
Age for per 1-year increase	1.03 (1.002, 1.06)	0.04
BMI for per 1-unit increase	1.22 (1.09, 1.37)	0.0004

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
