# Peer review of "Leptin, Ghrelin, and Leptin/Ghrelin Ratio in Critically Ill Patients"

_nutrients, 2019, doi:10.3390/nu12010036_

Round 1
Reviewer 1 Report
well written manuscript.
No specific comments of this article.
Author Response
Reply
Thank you for acknowledging our efforts
Reviewer 2 Report
Many thanks for the opportunity to review this manuscript. The authors correctly indicate in their discussion that one of the potential drawbacks of this study is the relatively small patient number however, despite this, the work provides some interesting results that warrant discussion. Please see some comments below:
Introduction
This section provides a useful overview of a complicated area however I was left wondering whether leptin/ghrelin response/ratio + a number of other inflammatory markers would depend on the reason for the patient being in ICU in the first place? Would underlying pathology (i.e. whether gastrointestinal in nature as an example) potentially influence an individuals ghrelin levels in particular?Methods
If using the multiplex kit from Millipore, presumably the authors measured "active ghrelin"? If so, can this be stated throughout the manuscript please and can it be stated that necessary inhibitors were added following sample collection? In the statistical analysis section, the authors state they used mann-whitney U tests - presumably a test for normal distribution was undertaken? If so, can this be stated please? Figure 1 demonstrates leptin, ghrelin etc response in high and low patients on different days - presumably this data should have been analysed using a two factor ANOVA? This was not clear in the statistical analysis section - can this be clarified please and, if this wasn't used, could some discussion on why be provided?Author Response
Comments and Suggestions for Authors
Many thanks for the opportunity to review this manuscript. The authors correctly indicate in their discussion that one of the potential drawbacks of this study is the relatively small patient number however, despite this, the work provides some interesting results that warrant discussion.
Reply
Thank you
Comment 1
Introduction
This section provides a useful overview of a complicated area however I was left wondering whether leptin/ghrelin response/ratio + a number of other inflammatory markers would depend on the reason for the patient being in ICU in the first place? Would underlying pathology (i.e. whether gastrointestinal in nature as an example) potentially influence an individual’s ghrelin levels in particular?
Reply
Thank you very much for this point. Unfortunately, the sample size does not allow subgroup analysis. We added this point now to the study limitations as follows:
“Because of the sample size, we were unable to perform subgroup analyses.”
Comment 2
Methods
If using the multiplex kit from Millipore, presumably the authors measured "active ghrelin"? If so, can this be stated throughout the manuscript please and can it be stated that necessary inhibitors were added following sample collection?
Reply
Thank you for this excellent observation. Yes, we used multiplex kit from Millipore to measure active ghrelin. We clarified this multiple times throughout the manuscript. In addition, we acknowledge that the serine protease inhibitor was not added to the blood samples after collection which has been added to the method as well as discussion section as limitation as follows;;
Methods:
“We did not add serine protease inhibitor to blood samples after collection”
Discussion:
“We did not add serine protease inhibitor to blood samples after collection, which may have resulted in underestimation of ghrelin levels”
Comment 3
In the statistical analysis section, the authors state they used mann-whitney U tests - presumably a test for normal distribution was undertaken? If so, can this be stated please?
Reply
Following sentence has been added to the statistical section as requested;
“Categorical variables were compared using chi-square or Fisher’s exact test and continuous variables using Mann-Whitney U test based on non- normality assumption and small sample size”
Comment 4
Figure 1 demonstrates leptin, ghrelin etc response in high and low patients on different days - presumably this data should have been analysed using a two factor ANOVA? This was not clear in the statistical analysis section - can this be clarified please and, if this wasn't used, could some discussion on why be provided?
Reply
We agree that it has not been made clear in the statistical method section. As such the following sentence was added;
“To assess whether low leptin/ghrelin ratio compared to high leptin/ghrelin ratio was associated with levels of different hormones and other inflammatory markers over time, we constructed repeated measure mixed linear model”
Reviewer 3 Report
In the present manuscript the authors studied different outcomes in patients in intensive care unit stratified based on the ratio leptin/ghrelin. However, no significant differences between groups were observed in the outcomes studied (only differences in fat mass due to the stratification by the ratio). In addition, the results obtained are discussed in the Results and Outcomes section, but the discussion does not have a section and is very short. Therefore, if the results obtained have a possible applicability or repercussion for these patients, the manuscript does not highlight it. In addition,
The authors refer in the abstract (line 19) and in the introduction (Line 46-47) as Icu patients, but it’s not explained, only is explained in the legend of Table 3 The quality of Figure 1 is poor and with errors such as “Adeponectin” and the units of some of the hormones studied should be changed to an appropriate scale (for example 1000000-10000000 pg/mL for Adiponectin is better ng/mL) Lane 60-70 “trial in which p critically ill patients” what means? In the Legend of the Figure 1. Box plots are displayed with…??Author Response
Comments and Suggestions for Authors
Comment 1
In the present manuscript the authors studied different outcomes in patients in intensive care unit stratified based on the ratio leptin/ghrelin. However, no significant differences between groups were observed in the outcomes studied (only differences in fat mass due to the stratification by the ratio). In addition, the results obtained are discussed in the Results and Outcomes section, but the discussion does not have a section and is very short. Therefore, if the results obtained have a possible applicability or repercussion for these patients, the manuscript does not highlight it.
Reply
Thank you for this comment. We have a detailed Discussion section, but we think during the editing process, the subtitle became unclear. We corrected this point.
Comment 2
The authors refer in the abstract (line 19) and in the introduction (Line 46-47) as ICU patients, but it’s not explained, only is explained in the legend of Table 3
Reply
Thank you for the suggestion. ICU has been replaced by its full form in both the abstract as well as the introduction as requested
Comment 3
The quality of Figure 1 is poor and with errors such as “Adeponectin” and the units of some of the hormones studied should be changed to an appropriate scale (for example 1000000-10000000 pg/mL for Adiponectin is better ng/mL)
Reply
Figure 1 has been replaced as suggested. Adeponectin has been replaced by “Adiponectin” In addition to this the units of adiponectin, adipsin, resistin and PAI-1 has been changed from pg/Ml to ng/mL.
Comment 4
Line 60-70 “trial in which p critically ill patients” what means? In the Legend of the Figure 1. Box plots are displayed with…??
Reply
Line 60 to 70, the letter p has been replaced by “the”. The legend of the Figure 1 has been completed to
“Box plots are displayed with medians and quartiles 1 and 3. The error bars refer to 10th and 90th percentiles”
Round 2
Reviewer 3 Report
After the revision of the manuscript, the style of this has been greatly improved. The authors have studied different outcomes in patients in intensive care unit stratified based on the ratio leptin/ghrelin. However, no significant differences were observed between the groups in the main results studied in the patients. The main conclusion of the manuscript is that the age and body mass index were significant independent predictors of high leptin-ghrelin ratio. Nevertheless, this result was to be expected because leptin is an adipokine proportional to fat mass, therefore, stratifying by a parameter containing leptin levels is similar to stratifying by fat mass. For these reasons, the interest of the manuscript and/or the Scientific Soundness in my view is low and my recommendation is to reject the manuscript.